# Trends in Telecare Use among Community-Dwelling Older Adults: A Scoping Review

**DOI:** 10.3390/ijerph192416672

**Published:** 2022-12-12

**Authors:** Nilufer Korkmaz Yaylagul, Hande Kirisik, Joana Bernardo, Carina Dantas, Willeke van Staalduinen, Maddalena Illario, Vincenzo De Luca, João Apóstolo, Rosa Silva

**Affiliations:** 1Department of Gerontology, Faculty of Health Sciences, University of Akdeniz, Antalya 07070, Turkey; 2Elderly Care Program, Vocational School of Haymana, University of Ankara, Ankara 06860, Turkey; 3Health Sciences Research Unit: Nursing (UICISA: E), Nursing School of Coimbra (ESEnfC), 3000-076 Coimbra, Portugal; 4SHINE 2Europe, 3030-163 Coimbra, Portugal; 5AFEdemy—Academy on Age-Friendly Environments in Europe, 2806 ED Gouda, The Netherlands; 6Dipartimento di Sanità Pubblica, Università degli Studi di Napoli Federico II, 80131 Napoli, Italy; 7Portugal Centre for Evidence Based Practice: A JBI Centre of Excellence (PCEBP), 3000-232 Coimbra, Portugal

**Keywords:** telecare, telehealth, telemedicine, older adults, home

## Abstract

A scoping review was conducted to map and analyze the concept of telecare services and the trends in telecare use. This scoping review was conducted according to Arksey and O’Malley’s framework. A search was conducted in CINAHL (via EBSCO), ERIC, Academic Search Ultimate, and MEDLINE/PubMed databases. This scoping review considered quantitative (e.g., analytical observational studies, including prospective and retrospective cohort studies, case-control, analytical cross-sectional, and descriptive-observational studies), qualitative (e.g., phenomenology, grounded theory, ethnography, and action research), and mixed-method primary studies. Forty research articles published from 1 January 2012, to 1 January 2022 were included in this review, these studies met the eligibility criteria as all were focused on telecare and targeting older adults over 65 living at home. The reviewers coded the data in an Excel spreadsheet, including the articles’ title, year, author, journal information and subject, research methods, sample size, location, and summary. Then, the researchers analyzed the conceptual definitions, measurement techniques, and findings in detail and the findings were grouped into categories. The trends around the concept of telecare are independent living, remote care, aging in place, and safety. Telecare research focuses mainly on service use, chronic illness, ethics, and cost-effectiveness. Technology acceptance among older individuals is a critical factor for telecare use. The results found in the literature about the cost-effectiveness of telecare are inconsistent.

## 1. Introduction

For many years, aging in place has been shown to support older people in maintaining their independence. However, the home environment is far from being ideal when it comes to caring for older adults, due to age-related conditions and changes such as falls, sensory loss, inactivity, multiple drug use, isolation, and loneliness. [1,2,3,4]. Moreover, institutional care is not enough to meet the growing needs of an aging society. Therefore, cost-effective complementary care alternatives are needed to support aging in place [5,6,7]. In this context, telecare technology is presented as an efficient and cost-effective solution [6,7].

The content of telecare services varies and so do their definitions. The United Kingdom’s Department of Health and Social Care (2009) defines telecare as “a service that uses a combination of alarms, sensors, and other equipment to help people live independently” [7]. According to the Scottish Government (2009), “telecare usually refers to equipment and detectors that provide continuous, automatic and remote monitoring of care needs, emergencies, and lifestyle changes, using information and communication technology” [8]. Peeters et al. [9] define telecare for older individuals as a technology-based service model consisting of medical care, outreach services, advice, monitoring, social and emotional support, security alarms, digital shopping services, educational and leisure activities, and services that support social contacts. Based on the aspects common to these definitions, telecare systems can be managed remotely and integrated into home care and other services. Telecare services involve monitoring patients’ health at a distance using technological solutions such as smartphones, audio or video equipment, and internet connection. Patients can be monitored through virtual consultations, video conferences, phone calls, and text messaging [10].

Due to the widespread use of telecare, mapping and organizing this information will be of interest to practice. For example, Ferreira Santana et al. [10] analyzed the use of telecare as a nursing intervention in caring for older people with Alzheimer’s disease and their caregivers. Stewart and McKinstry [11] assessed the association between older people’s fear of falling and telecare use, and examined whether telecare could reduce this fear of falling. In Scotland, Eccles [12] reviewed telecare technologies and examined the ethical issues raised by the telecare program. In a systematic review, Viana et al. [13] assessed the effectiveness of psychological, telecare, and educational interventions in improving adherence to treatment in patients with Type-1 diabetes. Moreover, Vitacca, Montini, and Comini [14] focused on how telecare can change clinical practice in patients with Chronic Obstructive Pulmonary Disease (COPD). Martínez-Alcalá et al. [15] conducted a systematic review of the literature on information communication technology (ICT) applications developed to assist patients with Alzheimer’s disease and their primary caregivers. However, a literature search found no studies on the state of the art of research on telecare services for older adults. Due to the broad scope and definition of telecare, we aimed to map and analyze the concept and trends of telecare.

## 2. Materials and Methods

Scoping reviews is a literature search technique that aims to quickly map the key concepts, main sources, and types of evidence supporting an area of research, particularly where a field is complex, or has not been extensively reviewed before [16]. Arksey and O’Malley’s scoping review model was used in this study. Arksey and O’Malley [16] suggest the following six steps for conducting scoping reviews: (1) identifying the research question, (2) identifying relevant studies, (3) study selection, (4) charting the data, (5) collating, summarizing, and reporting the results, and (6) consultation.

### 2.1. Identifying the Research Question

The following research question was addressed in this review: (1) What are the concepts and trends of telecare for community-dwelling older adults?

### 2.2. Identifying Relevant Studies

A search was conducted on CINAHL (via EBSCO), ERIC, Academic Search Ultimate, and PubMed/MEDLINE databases, frequently used in health, care, and aging studies. The Google Scholar search engine was used to optimize the search results of electronic database searches and improve the reliability of the search strategy. The keywords/text words in the titles and abstracts of relevant articles, and the index terms used to describe the articles, were used to develop a full search strategy. The main keywords used were telecare or telemedicine, or telehealth and older adults or elderly or geriatric, or geriatrics or aging or senior or seniors or older people or aged 65 or 65+ AND home. These keywords and other index terms were combined to search the databases.

This review considered studies published in English between 1 January 2012, and 1 January 2022. The authors believe that this timeframe will provide sufficient time to examine the trends of telecare research, due to the rapid technological development over this period. It also considered studies focused on telecare, telemedicine, and telehealth, involving community-dwelling older adults.

Thus, the inclusion criteria were established using the participants, concept, and context (PCC) framework: (P) Population: studies including participants (older adults) who receive a telecare service; (C) Concept: studies focusing on telecare or telemedicine; and (c) Context: studies conducted in the community, that is, studies in which telecare was delivered/implemented in the home environment.

Studies on the design, creation, setup, development, functionality, and promotion of the product or concept proof were excluded. The following exclusion criteria were applied: experimental studies on the functionality, setup, development, and promotion of technological equipment and medicine.

This scoping review considered quantitative (e.g., analytical observational studies, including prospective and retrospective cohort studies, case-control studies, analytical cross-sectional studies, descriptive-observational studies, including case series, individual case reports, and descriptive cross-sectional studies), qualitative (e.g., phenomenology, grounded theory, ethnography, and action research), and mixed-method primary studies.

### 2.3. Study Selection

A search was conducted on CINAHL (via EBSCO), ERIC, Academic Search Ultimate, and PubMed/MEDLINE databases. First, duplicates were removed. Two independent reviewers screened the titles and abstracts for eligibility against the inclusion criteria. Studies were selected based on the inclusion criteria. Reasons for excluding full-text papers that did not meet the inclusion criteria were recorded.

Thus, the search identified 3107 articles. A total of 825 duplicates were removed. The titles and abstracts were then screened for relevance (by NKY and HK). Of the 65 articles selected for full-text review (by NKY, HK, and JB), 40 studies were included in this scoping review. This process is detailed in Figure 1.

Any disagreements between the reviewers at each stage of the selection process were resolved through discussion or with a third reviewer (NKY, HK, RS, and JB). 

### 2.4. Charting the Data

The full texts of eligible studies were retrieved. Two independent reviewers (NKY and HK) assessed them in detail against the inclusion criteria. Studies that did not meet the inclusion criteria were excluded. Any disagreements that arose between the reviewers were resolved through discussion or with the intervention of a third reviewer (RS) at each stage of the study selection process. Finally, two independent reviewers (NKY and RS) extracted data from the studies using the methodology proposed by the Arksey and O’Malley’s [16] and an extraction tool developed specifically for this scoping review, based on the review objective and question. The reviewers (NKY and RS) coded the data in an Excel spreadsheet, including the articles’ title, year, author, journal information and subject, research methods, sample size, location, and summary. Then, the researchers (NKY, HK/JB, and RS) analyzed the conceptual definitions, measurement techniques, and findings in detail and the findings were grouped into categories. Table 1 includes each study’s author, year, country, participants, the concept of telecare services, trends of research on telecare, and research methodology.

### 2.5. Collating, Summarizing and Reporting the Results

A qualitative content analysis of concepts and research trends on telecare for community-dwelling older adults was conducted to examine key themes and their characteristics. In this context, open and axial coding types were used in this study. While in open coding, themes are identified based on the information, in axial coding, the sub-themes are identified and matched under relevant themes [18], by using the inductive process. The researchers independently summarized and tabulated the findings and reached a consensus on the table’s contents.

**Table 1 ijerph-19-16672-t001:** Descriptions of research articles included in the scoping review on “Telecare”.

Author, Year, Country	Study Population	Concept of Telecare Services’	Trends of Research on Telecare Use	Study Method
Watson, Bearpark and Ling (2020) [7], England	*n* = 23 (Service users commissioners service providers caregivers)	This study explored the impact of these rapid response and telecare services on service users and their carers. It also investigated how telecare was viewed by commissioners, including adult social care services and other stakeholders.	**Service Use**—The use of a combined rapid response and telecare service resulted in older people remaining independent in their homes for longer, which improved their reported quality of life and relieved stress on carers and pressures on other service providers.	Qualitative research (semi-structured interview)
Chang et al. (2013) [1], Taiwan	*n* = 13 (Nurses experienced in telecare)	Advances in science, technology, and healthcare have contributed to the global growth of an aging population. With a concomitant decline in birth rate, the increasing need for elder care cannot be met effectively by traditional care models as fewer caregivers are available. Data contents were analyzed to explore nurses’ perceptions of telecare services. Five major themes were identified: (1) provision of individualized care, (2) increasing job requirements and stress, (3) working with a collaborative care model, (4) understanding concerns of the elderly, and (5) foreseeing future challenges.	**Service Use**—The participants identified the advantages of telecare services for older people and recognized a new opportunity for health management in the future population and social changes. Using technology to provide telecare may offer an effective supplement for elder care services.	Qualitative research (semi-structured interview)
Millan-Calenti et al. (2017) [19], Spain	*n* = 742 (65+ who use a first-generation telecare service)	Telecare is a healthcare resource based on new technologies that through the services offered, help older people to continue living in their homes. In this sense, first-generation telecare services have quickly developed in Europe. This study aimed to define the profile, pattern of medication consumption and disease frequencies of elderly users of a telecare service.	**Service Use**—Regardless of the social elements contributing to the implementation of telecare services, specific health characteristics of potential users, such as morbidity and polypharmacy, should be carefully considered when implementing telecare services in the coming years.	Quantitative research (survey)
Hamblin (2017) [20], United Kingdom	*n* = 60 (Telecare service users aged 65+)	This paper explored the factors influencing older adults’ telecare acceptance and optimal use. This paper uses data collected by a qualitative, multi-method, longitudinal research study to explore whether an American model of ‘obtrusiveness’ is applicable to the United Kingdom’s context by examining what factors influence older adults’ acceptance and use of telecare.	**Service Use**—Telecare is a key part of health and social care arrangements, and consideration of issues related to the obtrusiveness model outlined and explored empirically in this study will help professionals ensure devices are accepted and used effectively. In addition, two further issues which affect the uptake and use of telecare—the degree of control a person feels they have over their health and social care arrangements, and the information and support they receive in using telecare—emerged from the data.	Qualitative research (observation, informal conversation, semi-structured interview)
Wherton et al. (2015) [21], England	*n* = 61 (Users of telehealth and telecare, their carers, service providers and technology suppliers)	The low uptake of telecare and telehealth services by older people may be explained by the limited involvement of users in the design. If the ambition of ‘care closer to home’ is to be realized, then industry, health and social care providers must evolve ways to work with older people to co-produce useful and usable solutions.	**Service Use**—Analysis revealed four main themes. First, there is a need to raise awareness and provide information to potential users of assisted living technologies (ALTs). Second, technologies must be highly customizable and adaptable to accommodate the multiple and changing needs of different users. Third, the service must align closely with the individual’s wider social support network. Finally, the service must support a high degree of information sharing and coordination.	Qualitative research (ethnographic case study)
Mort et al. (2013) [22], England	*n* = 166 (Older individuals, informal and formal caregivers, older individuals in independent assisted accommodation)	Telecare and telehealth developments have recently attracted much attention in research and service development contexts, where their evaluation has predominantly concerned effectiveness and efficiency. Their social and ethical implications, in contrast, have received little scrutiny. This study aimed to develop an ethical framework for telecare systems based on analysis of observations of telecare-in-use and citizens’ panel deliberations.	**Ethics**—Telecare has care limitations; it is not a solution, but a shift in networks of relations and responsibilities. Telecare cannot be meaningfully evaluated as an entity, but rather in the situated relations people and technologies create together. Characteristics of ethical telecare include on-going user/carer engagement in decision making about systems: in-home system evolution with feedback opportunities built into implementation.	Qualitative research (ethnographic study, observation interview, data from panels and participative conferences)
Steils et al. (2021) [23], England	Quantitative research: *n* = 152 (local council members providing telecare services in England Qualitative research: *n* = 25 (Qualitative interviews were conducted with telecare managers of 25 local councils)	Telecare has become a ubiquitous part of social care offered by English local council adult social care departments, whether directly provided or commissioned from external partners. Outcomes of telecare use for older people have been extensively researched and studies of family and other unpaid/informal carers and telecare use have suggested that carers find that it offers reassurance.	**Service Use**—The telecare can support carers in their role, e.g., enabling them to have respite from caring or to engage in other activities, and, in some cases, remain in employment. It also confirms others’ findings that carers are often essential for telecare to work for older people, whether by ensuring equipment is functioning correctly or responding to alerts. In addition, when telecare does not meet the needs and goals of the older person, carers initiate changes to equipment or services, or request the removal of telecare.	Combined research (Quantitative research, survey and qualitative research (semi-structured interview)
Maclnnes (2020) [24], England	*n* = 7 (Care navigators)	Telecare is seen as a potential means of addressing the future care needs of aging societies. Care navigators and other health and social care support workers are increasingly responsible for prescribing telecare.	**Service Use**—Care navigators are well-placed to prescribe telecare. There is a need for training which addresses knowledge and skills related to decision-making. An investment should be made in peer support networks.	Qualitative research (semi-structured interview)
Bentley et al. (2018) [25], England	*n* = 22 (65+, who are known to refuse to join their local telecare agency)	Despite the reported benefits of telecare use among older adults, uptake of telecare in the United Kingdom remains relatively low. Non-users of telecare are an under-researched group in the telecare field. Twenty-two qualitative individual semi-structured interviews were conducted to explore the views and opinions of current telecare non-users regarding barriers and facilitators to its use, and examine considerations that may precede their decision to accept, or reject, telecare.	**Service Use**—A cost-benefit decision process appears to take place for the potential user, whereby the benefit of peace of mind is weighed against the perceived ‘costs’ of using telecare. Telecare is often perceived as a last resort rather than a preventative measure. A number of barriers to using telecare need to be addressed if individuals are to make fully informed decisions about their use of telecare, and to begin using telecare at a time when it could provide them with optimal benefits.	Qualitative research (semi-structured interview)
Cook et al. (2018) [26], United Kingdom	*n* = 14 (Family caregivers of older individuals receiving telecare services)	In the United Kingdom an aging population met with reduced funding for social care has led to a reduction in support for the older people marked with an increased demand on family caregivers. Assistive telecare devices are seen as an innovative and effective way to support older people. However, there is limited research that has explored the adoption of telecare devices from the perspectives of family caregivers.	**Service Use**—Assistive telecare devices were viewed positively, considered easy to use, useful and functional, with ensuring patient safety being one of the main reasons for adoption. Efforts to increase adoption and engagement should adapt recruitment strategies and service pathways to support both the patient and their caregiver.	Qualitative research (semi-structured interview)
Chou et al. (2012) [27], Taiwan	*n* = 30 (Primary caregivers of dementia patients using telecare)	The telecare medical support system has been a part of dementia care in many countries for many years. Although worth considering, the telecare medical support system model is difficult to directly implement in Taiwan because of cultural and social issues.	**Chronic Diseases**—The telecare medical support system as an effective tool that helps reduce primary caregiver isolation and uncertainty and provides 24-hour care management and safety checks using advanced technology and a professional care team. The telecare medical support system can effectively improve dementia care.	Qualitative research (semi-structured interview)
Johannessen et al. (2019) [28], Norway	*n* = 10 (Home care professionals working at municipal home care centers)	The use of telecare technology has the potential to maintain and improve older adults’ independence and quality of life, reduce hospital and care home admissions, and allow them to remain in their own homes for longer. The objective of this study was to explore home care professionals’ perceptions of safety related to the use of telecare by older adults.	**Service Use**—The use of telecare protects older adults, giving them a sense of security. However, they also stated that the use of telecare involves challenges that can lead to harm to older adults, due to technological limitations and difficulties in managing and understanding the technology. Although telecare can improve safety, it is necessary to develop reliable technology and adapt it to the user’s skills, abilities, and resource.	Qualitative research (focus group study)
Woolham et al. (2021) [29], England	*n* = 152 (Social workers, care managers, and other professionals)	This article explores the role of telecare assessment, review and staff training in meeting the needs of older people living at home. How telecare is used rather than telecare itself shapes outcomes for the people who use it, and “sub-ideal” telecare outcomes may be linked to how telecare is adopted, adapted and used. This is influenced by employee training and telecare availability and does not consider telecare as a complex intervention.	**Service Use**—In using telecare to achieve the best outcomes for older people, social workers, care managers, and other professionals involved in telecare assessment will need enhanced training opportunities, and their employers will need to perceive telecare as a complex intervention rather than simply a ‘plug and play’ solution.	Quantitative research (survey)
Delgado and Paschoarelli (2020) [30], Brazil	*n* = 30 (Older people using telecare services)	The definition of telecare focuses on the ability of the service to provide assistance, essentially social or medical, in any place and in any situation, regardless of the technologies used. This study aimed to determine positive and negative factors experienced by older people in the use of a Remote Care Application: the Me Cuido App. The older people’s experiences with the use of the application can further contribute to the implementation of new assistance technologies related to telecare in the Brazilian population.	**Service Use**—The results showed that older people in Brazil have different ways of understanding ageing at home with Telecare technologies. In this sense, different repertoires have been identified on the positive and negative factors for the implementation of a Remote Care Application for older people living alone in Brazil, for example security, independence, peace of mind for the family, older person’s privacy, social differences in Brazil, and app design and usability.	Qualitative research (focus groups)
Etemad-Sajadi and Dos Santos (2021) [31], Switzerland	*n* = 213 (Older people using connected health technologies (assistive alarm, telecare, sensors, etc.) at home and receiving health care at home.)	Connected technologies cannot prevent transfer to a nursing home but may enable older people to stay in their houses for longer. The goal of this study is to focus on the feeling of social presence (the perception that there is personal human contact) with these connected technologies and the degree of trust of elderly people using these systems and identify whether both aspects can have an impact or not on the quality of service delivered by home care service companies.	**Service Use**—The positive influences of the feeling of social presence and trust on the quality of service perceived show that the use of connected health technologies can impact positively the overall perception of the service quality. It was also found that people who live alone are more willing to accept the use of connected health technologies compared to people who live with a husband or wife. In addition, it was observed that the more advanced in age older people are, the more they judge that these technologies positively impact connection to the outside world.	Quantitative research (survey)
Moo et al. (2020) [32], USA	*n* = 230 (Dementia clinic outpatients and their families)	People with dementia face barriers to attending face-to-face medical care. Despite the potential of video telemedicine to alleviate these barriers, little is known about video telemedicine at home for dementia.	**Chronic Diseases**—In this study, 96% of participants agreed to join video telemedicine, or gave reasons for declining, with the main reasons for participating being convenience and less disruption of routines. Thus, people with dementia and their families were willing to enroll in a home telemedicine clinic. Satisfaction with home visits was high and equal to visits at the clinic.	Quantitative research (survey)
Bjørkquist et al. (2019) [33], Norway	*n* = 7 (Primary health care service managers)	As an increasing number of seniors with complex needs are living at home, the implementation of telecare has become a priority. This article aims to identify factors that influence inter- and intra-organizational collaboration in Norwegian primary care. The focus is on collaboration in providing services to senior users with telecare solutions, in this case personal alarms.	**Service Use**—Challenges and barriers to collaboration and integration were information flow and information sharing, unclear understanding of the division of functions between the units involved and their employees, and the lack of meeting points between the emergency medical center and homecare. The introduction of telecare does not simplify collaboration or improve services; technology does not solve collaboration challenges. Technology limits information to written form, which may not meet service provider and user needs. Collaboration and integration require common strategies and leadership that implement them, including in telecare.	Qualitative research (semi-structured interview)
Lyth et al. (2019) [34], Sweden	*n* = 94 (Older patients with chronic obstructive pulmonary disease or heart failure)	Growing populations of elderly patients with chronic obstructive pulmonary disease (COPD) or heart failure (HF) require more healthcare. A four-year telehealth intervention was implemented to assess whether patients with advanced COPD or HF would have lower hospitalization rates when using the telehealth intervention. The objective was to investigate the effects of the intervention on healthcare costs and the number of hospitalizations.	**Chronic Diseases/Cost-effectiveness**—The results show that the intervention using an active telemonitoring system was effective for both HF and COPD patients in reducing hospitalizations of patients who died during the study period, as well as for patients who survived the entire study period. In addition, the intervention reduced the number of visits to primary care, emergency care and other outpatient care.	Quantitative research (cohort study)
Wade et al. (2016) [35], Australia	*n* = 19 (Senior clinicians, health service managers and policy makers)	Telehealth can be used to provide specialized services at homes, such as rehabilitation or palliative care, that would otherwise be provided in the hospital or in face-to-face home visits. This study tested combining home telehealth with the existing rehabilitation, palliative care, and geriatric outreach services. Due to the known difficulty in transitioning telehealth project services, a qualitative study was carried out to produce a preferred implementation approach for sustainable and, large-scale operations, and a process model offering practical advice to achieve this goal.	**Service Use**—A change management model for transitioning a home telehealth project from a trial to a routine service was built from scratch. Key components of this model, suggested by the qualitative data analysis, were new clinical and business models of care and evidence of benefits, which in turn supported clinician acceptance.	Qualitative research (semi-structured interview)
Kalicki et al. (2021) [36], USA	*n* = 16 (Primary care physicians)	Video-based telehealth has emerged as an important innovation in care delivery within home-based primary care. Not only does telehealth reduce patient costs, transportation and time, but the COVID-19 pandemic has also highlighted additional benefits, such as reduced exposures to infections. The main objective was to identify the main barriers to the use of video-based telehealth among homebound older adults.	**Service Use**—The COVID-19 pandemic has resulted in a major and dramatic shift to the use of video-based telehealth in home-based primary care. Patients without caregiver support to assist with the technology may benefit from new approaches, such as deploying community health workers to assist with device setup. Clinicians may not be able to identify potentially modifiable barriers to telehealth use among their patients, highlighting the need for better systematic data collection prior to targeted interventions to increase video-based telehealth use.	Quantitative research (survey)
Hawley et al. (2020) [37], England	*n* = 50 (Older patients with scheduled clinic visits)	COVID-19 has exacerbated the problem of accessing care due to reduced clinic visits, transport restrictions and other social measures to mitigate the pandemic. Professionals are looking for ways to provide effective virtual care for older patients in line with regulations to reduce the spread of COVID-19. Thus, the objective was to identify and address barriers perceived by the patient to integrate telehealth home visits.	**Service Use**—They categorized patients into four phenotypes based on their interest and capability to complete a home telehealth visit: interested and capable, interested and incapable, uninterested and capable, and uninterested and incapable. These phenotypes made it possible to create trainings to overcome barriers perceived by the patients. Thirty-two home telehealth visits and 12 post-visit interviews were carried out. Formative assessment showed that the pilot was successful in overcoming many barriers perceived by the patients. All respondents reported that home telehealth visits improved their well-being.	Mixed research (Exploratory sequential)
Kim et al. (2019) [38], USA	*n* = 516 (Homecare staff)	Despite the increasing evidence for the effectiveness of telehealth technology in screening and treating depression in older adults, they have been slowly adopted by Home Health Care (HHC) agencies. Therefore, this study was conducted to determine how HHC agencies perceive and use telehealth technology for depression care among homebound older adult patients.	**Service Use**—The majority of the staff had a neutral or positive perception towards telehealth. Factors such as fewer years of experience in using telehealth and a small annual budget may explain a negative perception towards telehealth. Therefore, further education and resources are needed to support telehealth use. Future studies may consider comparing telehealth programs and identifying supporting policies.	Quantitative research (survey)
de Luca et al. (2021) [39], Italy	*n* = 60 (Frail older people)	Aging and age-related issues are affecting healthcare systems worldwide, so the use of telemedicine offers a possible solution.	**Chronic Disease**—Telemedicine can be considered an important tool to improve the psychological health and quality of life of frail older patients living at home.	Quantitative research (randomized controlled study)
Lai et al. (2020) [40], Hong Kong	*n* = 60 (Older adults with neurocognitive disorder [NCD] and their caregivers)	Social distancing under the COVID-19 pandemic has restricted access to community services for older people with NCD and their caregivers. Telehealth is a viable alternative to face-to-face services. Telephone calls alone, however, may be insufficient. This study aimed to assess whether supplementary telehealth through video conferencing platforms can bring additional benefits to patients with NCD and their spousal caregivers at home.	**Service Use**—Telemedicine was associated with improved resilience and well-being for both people with NCD and their caregivers at home. The benefits were already visible after 4 weeks and unmatched by telephone alone. Video conference as the telemedicine modus operandi, beyond the context of social distancing related to the pandemic, must be considered.	Quantitative research (Interventions/pretest−post-test design)
de Cola et al. (2020) [41], Italy	*n* = 131 (Older individuals aged over 65 living alone or spending most of the days alone at home)	Scientific advances and new information and communication technologies have facilitated the development of services that allow older people to stay at home as long as possible. This study assessed the usability and patient satisfaction of a new telemedicine system. All enrolled participants participated in a telecare program, which included remote surveillance services and teleconsultation with different health professionals, including nurses.	**Service Use**—Telemedicine has proven to be useful in improving health and quality of life of disadvantaged older people, especially if affected by severe comorbidity and living far from health services. Moreover, the patient satisfaction concerning the service was rated as good by the majority of the participants, although the usability rate was not so high.	Quantitative research (survey)
Norman et al. (2018) [42], USA	*n* = 7 (Home-based medical care provider)	Home-based primary care is a multidisciplinary, ongoing care strategy that can provide cost-effective home care to meet the needs of homebound, medically complex older adults in the USA. This study assessed home-based primary care practice from six sites to better understand the different operation structures, common challenges, and approaches to providing home-based primary care.	**Service Use**—The aggregated case studies revealed important issues focused on team composition, patient characteristics, use of technology and urgent care delivery. Although many practices offered urgent care, practices varied in the methods used to provide care including the use of community paramedics and telehealth technology	Qualitative research (unstructured interview)
Ladin et al. (2021) [43], England	*n* = 60 (Clinicians, older patients, carers)	Telehealth has been posited as a cost-effective means for improving access to care for persons with chronic conditions, including kidney disease. Perceptions of telehealth among older patients with chronic illness, their care partners, and clinicians are largely unknown but are critical to successful telehealth use and expansion efforts. This study aimed to identify patient, care partner, and nephrologists’ perceptions of the patient-centeredness, benefits, and drawbacks of telehealth compared to in-person visits.	**Cost-effectiveness/Chronic Disease**—Older patients, care partners, and kidney clinicians (i.e., nephrologists and physician assistants) shared divergent views of patient-centered telehealth care, especially its clinical effectiveness, patient experience, access to care, and clinician-patient relationship.	Qualitative research (semi-structured interview)
Shah et al. (2013) [44], USA	*n* = 21 (Patients, family caregivers, telemedicine dispatcher, certified telemedicine assistants, telemedicine providers, and senior living communities’ staff.)	Telemedicine has the potential to enhance the provision of emergency care for older adults. Using unconstrained qualitative methods, this study aimed to understand the opinions of participants, caregivers and health care providers participating in a program that provided telemedicine-enhanced emergency care to older adults living in assisted and independent living communities.	**Service Use**—Stakeholders find the convenience and speed of telemedicine-enhanced emergency care to be highly desirable. Providers felt that telemedicine-enhanced emergency care provided sufficient data, better diagnostic certainty, and overall improved care, although they felt that face-to-face visits were superior. Barriers related to training and technology require special attention.	Qualitative research (semi-structured interview)
Kim et al. (2018) [45], USA	*n* = 20 (Staff of home health care agencies that are members of the National Association for Home Care & Hospice (NAHC) in the USA)	Despite growing evidence for the effectiveness of telehealth technology in screening and treating chronic illness, and comorbid depression among older adults, it has been slowly adopted by home health care agencies. This study aimed to identify factors that determine the adoption of telehealth technology.	**Cost-efetiviness**—Most participants perceived telehealth as effective for symptom management and cost reduction. Meanwhile, some participants had mixed feelings about telehealth for depression care, as they did not recognize their agency as equipped with the necessary resources and trained personnel. In addition, significant determinants of telehealth adoption included agency-related characteristics, patient-home environment, reimbursement and cost-related factors, and employees’ perception of telehealth.	Mixed-methods research (Quantitative research, online survey. Qualitative research, in-depth interview)
Naick (2018) [46], England	*n* = 11 (Care navigators and managers)	The provision of telecare for the older adults in England is increasingly being facilitated by care navigators in the non-statutory sector. The purpose of this article is to explore the experiences of care navigators in the assessment of the older adults for telecare and to understand what contextual and organizational factors impact their practice.	**Service Use**—Findings suggest that strategic placement of care navigators could support the demand for telecare assessment to facilitate hospital discharges. This study highlights the perception of home assessment as the gold standard of practice for care navigators. To develop a more sustainable model for care navigators’ capacity to work in hospital teams and provide home assessments needs to be further explored.	Qualitative research (semi-structured interview)
Taylor et al. (2015) [47], Australia	*n* = 180 (Hospital-based clinicians, therapists, nurses, and doctors)	The study was an action research initiative that introduced and evaluated the impact of telehealth services on palliative care patients living in the community, home-based rehabilitation services for the elderly, and services to the elderly in residential aged care. The aim of this study was to understand the issues encountered during the provision of technology services that supported this trial.	**Cost-effectiveness**—The efficient management of consumer devices in multiple settings will become critical as telehealth services grow in scale. Effective collaboration between clinical and technical stakeholders and further workforce education in telehealth can be key enablers for the transition of face-to-face care to a telehealth mode of delivery.	Mixed methods research (Action research)
Solli and Hvalvik (2019) [48], Norway	*n* = 6 (Nurses)	In Norway, changes in life expectancy have led to increased attention to older people who are aging at home, by means of home care services, adapted technology and informal caregivers. The caring situation has become difficult for many caregivers. The use of telecare has now offered them the possibility to receive support at home. The purpose of this study was to explore how nurses provide support and care at a distance, using a web camera and a web forum in a closed telecare network for caregivers to persons suffering from stroke and dementia.	**Service Use**—The nurses provided long distance support and care for the caregivers, by using computer-meditated communication. This communication was characterized by closeness as well as empathy. To strengthen the caregivers’ competence and independence, the nurses were easily accessible and provided virtual supervision and support. This study contributes to knowledge about balancing in the relationship, as well as knowledge about bridging the gap between technologies and nursing care as potential conflicting dimensions.	Qualitative research (semi-structured interview)
Betkus et al. (2020) [49], Canada	*n* = 95 (Consultation letters)	Telehealth has the potential to support the care of older adults and their desire to age at home by providing a videoconferencing connection to specialist geriatric care. However, more information is needed to determine how telehealth services affect the care of older adults, and how telehealth services for older adults compare to traditional in-person methods of care provision. The aim of this study was to compare telegeriatric and in-person geriatric consultation methods with respect to outcomes and costs.	**Cost-effetiviness**—Telehealth consultations cost substantially less than in-person consultations and are a promising way to improve access to geriatric care for older adults in underserved areas.	Qualitative research (retrospective study)
Sánchez Criado and Domènech (2015) [50], Spain	*n* = 103 (Engineers, policymakers, service managers, technicians and telecare users)	This paper offers an ethnographic interpretation of how in a changing context of family care different Spanish home telecare services provide older people with social links to prevent their isolation, granting them “connected autonomy”: the promotion of their autonomy and independent living through connectedness.	**Ethics**—From this perspective, the greatest challenge telecare and other similar technological care practices face lies in how to transform the relational promise of an old age in “connected autonomy” into other figurations of “collectivised care”	Qualitative research (interview and focus group)
Steffen and Gant (2015) [51], USA	*n* = 74 (Women caring for an older adult with a neurocognitive disorde r[NCD])	This study analyzed the differential impact of two telehealth programs for women caring for an older adult person with a NCD. The outcomes examined were depressive symptoms, upset following disruptive behaviors, anxious and angry mood states, and caregiving self-efficacy.	**Chronic Diseases**—This study provides some initial evidence for the effectiveness of a telehealth behavioral coaching intervention compared to basic education and telephone support. Caregivers’ abilities to maintain strategy use during progressive disorders such as Alzheimer’s disease likely require a longer intervention contact than provided in this study. Dementia caregivers, including those living in rural areas, can benefit from affordable, empirically supported interventions that can be easily disseminated across distances at modest cost.	Quantitative research (randomized controlled study)
Reeder et al. (2013) [52], USA	*n* = 96 (Frail older adults)	Older people with multiple chronic conditions face the complex task of medication management involving multiple medications of varying doses at different times. Advances in telehealth technologies have resulted in home-based devices for managing medication and monitoring the health of older adults. This study examined older adults’ perceptions of a telehealth medication dispensing device as part of a clinical trial involving home health clients, nurse coordination, and use of a medication dispensing device.	**Service Use**—The technology-enhanced medication management device in this study is an acceptable tool for older adults to manage medication in collaboration with home care nurses.	Quantitative research (survey)
Gellis et al. (2012) [53], USA	*n* = 102 (control group = 51, intervention group = 51) (Older adults diagnosed with hearth failure [HF] or chronic obstructive pulmonary [COPD])	Telehealth is emerging as a viable intervention model to treat complex chronic conditions such as HF and COPD and to involve older adults self-care conditions.	**Chronic Diseases**—At follow-up, the telehealth intervention group reported greater increases in general health and social functioning, and improved depressive symptom scores compared with the usual care plus education group. The control group had significantly more emergency room visits than the telehealth group. There was a trend towards a lower number of hospital days among telehealth participants. Telehealth can be an efficient and effective method of systematically providing integrated care in the home health sector. The use of telehealth technology can benefit homebound older adults who have difficulty accessing care due to disability, transportation, or isolation.	Quantitative research (randomized controlled study)
Murphy (2018) [54], USA	*n* = 187 (Telehealth and medical records of older veterans)	Health monitoring within a telehealth program is a strategy to efficiently care for older people with heart failure (HF). Limited description is identified in the literature for the extent participant submissions trigger an alert or monitoring nurses transfer telehealth alert-range data to the medical record.	**Chronic Diseases**—Clinical relevance of alert-range telehealth data is uncertain partly due to frequent alerts triggered by physiological submissions, few changes in health status observed by the participant or monitoring nurse, and non-significant association between the proportion of alerts or nurse response, respectively, with baseline demographic or clinical measures.	Qualitative research (retrospective chart review)
Solli et al. (2015) [55], Norway	*n* = 15 (Nurses, caregivers)	In Norway and other European countries, there is a greater focus on aging at home, which is aided by technology as well as formal and informal care. With computer-mediated communication, such as telecare, it is possible for nurses to provide supportive care to caregivers in their homes. This study aimed to explore the relationship between nurses and caregivers using a web camera and a web forum as communication methods.	**Service Use**—Nurses dynamically responded to the information they received and helped to empower individual caregivers and strengthen interpersonal relationships between caregivers. While some participants found meeting in a virtual room to be close and intimate, others wanted to keep some distance.	Qualitative research (content analysis)
Cimperman et al. (2013) [56], Slovenia	*n* = 87 (Older adults)	The success of home telemedicine depends on end-user adoption, which has been slow despite rapid advances in technological development. This study focuses on an examination of significant factors that may predict successful adoption of home telemedicine services in older adults.	**Service Use**—The results reveal seven predictors that play an important role in the perceptions of home telemedicine services: perceived usefulness, effort expectancy, social influence, perceived security, computer anxiety, facilitating conditions, and physicians’ opinion. The results provide important insights into older adults’ home telemedicine acceptance, with guidelines for the strategic planning, development and commercialization of home telemedicine for the gray market.	Qualitative research (focus group)

### 2.6. Consultation

In addition to the literature review, the scoping review should also include a consultation exercise. For this purpose, two focus groups were conducted. The first one took place during the design phase to determine the review questions and the databases. The second one included a presentation of the research findings at the reporting stage and a discussion of the gaps and suggestions in the literature. The focus group consisted of a physician, an expert in telecare technologies, an older care specialist, a chronically ill patient using a telecare service at home, and an older adult living alone. The first focus group was conducted by NY with the collaboration of HK, and the second was conducted by HK with the collaboration of NY.

## 3. Results

### 3.1. Study Characteristics of the Study

The included studies were conducted in several countries. The UK had the highest number of studies (*n* = 12), followed by the USA (*n* = 10). Four studies were conducted in Norway. Taiwan, Spain, Italy, and Australia had two studies each, and Slovenia, Brazil, Hong Kong, Sweden, Switzerland, and Canada had only one study each. Concerning the timeframe, the majority of studies were published in 2020 (*n* = 9), followed by 2021 (*n* = 6) and 2018 (*n* = 6). Different research methodologies were used. Table 1 shows the characteristics and research objectives of the included articles.

### 3.2. Key Themes and Findings

The data obtained from the reviewed articles reveal that telecare research focuses on four key themes: service use (*n* = 27), chronic diseases (*n* = 6), cost-effectiveness/ chronic diseases (*n* = 2), cost-effectiveness (*n* = 3), and ethics (*n* = 2). Figure 2 shows the key themes and sub-themes of the included studies analyzed.

#### 3.2.1. Service Use

Studies on the users’ perspectives discussed the effects of telecare on caregivers and users, the factors influencing telecare use, users’ profiles, service perceptions and experiences, and telecare use and acceptance. Therefore, these studies revealed that caregivers’ well-being improved [23,26] and that telecare users remained longer in their homes, delaying their transfer to institutional care [7].

Older people’s technology acceptance level is critical for telecare use [36,56]. Cook et al. [26] demonstrate that family caregivers, rather than patients, are the main decision-makers regarding the use of telecare services. This is particularly true in cognitive diseases, where family caregivers are responsible for deciding on telecare use based on the patient’s best interests.

Moreover, Millan-Calenti et al. [19] revealed that the most frequently reported diseases among older adults using telecare services are hypertension, rheumatic disorders, and coronary artery disease.

There are some obstacles and facilitators associated with telecare use. The major factors negatively affecting the acceptance of telecare are stigma design, alternative options, and cost. Hawley et al. [37] argue that the most significant obstacles to telecare use are the complexity of technological equipment and the undermining of identity and independence. Using telecare can make individuals feel that they are not in control of their life, and it can increase domestic dependency [36,37]. On the other hand, factors such as advanced age, more physical and cognitive disabilities, and advice from close relatives also facilitate the acceptance of telecare [25,31,32].

Hamblin [20] emphasizes that some users considered telecare a restful option in terms of security and a threat to their identity as independent individuals, seeing it as stigmatizing and a source of social embarrassment. In addition, they tended to avoid using telecare devices if they did not know how they worked. Similar to the findings Hamblin [20] reported, this study also found that telecare can stigmatize older people. In line with other studies, they consider telecare a security measure against the dangerous external environment.

Telecare use was also addressed from the perspective of service providers. Nurses are the main providers of telecare services, so telecare patients turn to nurses to help them solve the problems they encounter. Therefore, nurses are expected to master this type of technology. Moreover, users’ service acceptance and continuity are ensured by a relationship of trust between patients and nurses in face-to-face home visits [1]. Another telecare service provider is the local government. Given that family caregivers should be well informed about telecare and its functions to use it successfully, more training should be provided on the use and adjustment of telecare devices and their functions [23]. In a study with homecare professionals, Johannessen et al. [28] found that telecare use prevents older individuals from being harmed and gives them a sense of safety. However, they also noted that telecare use involves challenges that may harm older adults due to technological limitations and difficulties in managing and understanding the technology. With care professionals, Woolham et al. [29] indicated that it is the way of using telecare rather than telecare itself that shapes the outcomes of telecare use. In the United Kingdom, care navigators are consulted in the decision-making process for prescribing telecare services to older people living at home. Therefore, care navigators also play a critical role in providing telecare services, assessing patients’ social characteristics, medical history, physical and cognitive abilities, and environmental factors during the decision-making process [24].

#### 3.2.2. Chronic Diseases

Telecare in patients with chronic diseases will reduce hospitalizations, increase patients’ quality of life, and reduce care staff workloads [27,32,34,53].

Telecare seems a feasible solution to monitor COPD patients and intervene rapidly in exacerbation processes. Lyth et al. [34] examined the level of acceptance of telemonitoring services among older adults with COPD and concluded that they significantly reduce hospitalization rates and healthcare costs.

Although there are various care models for dementia patients, telecare is a promising and feasible alternative to caring for these patients [32]. Chou et al. [27] showed that patients with dementia seek telecare support to solve sleep problems, obtain dietary and medical advice, and receive information about daycare services or additional support. For their caregivers, telecare ensures patient safety. Moreover, telecare use is more effective in patients with early-stage dementia than in those with advanced-stage dementia [27].

Gellis et al. [53] argue that using telehealth technology can benefit homebound older adults who have difficulty accessing care due to disability, transportation, or isolation.

In a study with frail older adults, de Luca et al. [39] found that telemonitoring improved their mood and nutritional status. Ladin et al. [43] have researched the advantages and disadvantages of telecare use in older individuals with chronic kidney diseases and found that besides being support in care participation, the inequalities in access of visual services were indicated as the disadvantage of telecare. Steffen and Gant [51] found that a telecare behavioral coaching intervention significantly reduced older people’s negative mood states.

#### 3.2.3. Ethics

Telecare can influence older people’s domestic processes and privacy and may impose additional responsibilities on caregivers. Therefore, it is crucial to look at telecare from an ethical perspective for service users, caregivers, service providers, and policymakers. The biomedical ethical framework is widely accepted in health. However, in telecare, social care issues come first before biomedical ethical principles. The policies made and the services provided are expected to be shaped based on social justice. Telecare is also considered a service reproducing macro-level social inequality, for example, in geographical, technological, economic, and educational areas. Although autonomy is a key component of telecare policies, the importance of the interdependence of human relations should be considered. In this respect, the use of telecare in reducing or increasing isolation among older people should also be discussed [12].

Further studies are needed to gain more insight into the ethical issues involved in telecare use. The search identified only two studies on the ethical framework of telecare. Mort et al. [22] called attention to the fact that, contrary to expectations, telecare leads to additional work and responsibilities for care recipients. They also emphasized that telecare is not designed to be easily used by older individuals and can reduce privacy and be a highly intrusive model. Sanchez Criado and Domenech [50] argue that telecare does not support older people’s full autonomy and that the fragmentation of care services leads to the invisibility of care and overburden.

#### 3.2.4. Cost-Effectiveness

New service models such as telecare, telemedicine, telehealth, and medical informatics applications have emerged as a manifestation of technological developments in the health system. At this point, it is critical to assess the cost-effectiveness and cost-benefits of these new service models to improve health outcomes, use scarce resources efficiently, and help healthcare decision-makers decide on their feasibility [57].

Although there are several cost-effectiveness studies on e-health, this review identified only four studies on telecare and aging. Kim et al. [45] reported key determinants of telehealth adoption, including agency-related characteristics, the patient-home environment, reimbursement and cost-related factors, and staff perceptions of telehealth.

Betkus et al. [49] revealed that telehealth geriatric consultations cost significantly less than in-person consultations and are a promising way to improve access to geriatric care for older adults in underserved areas.

According to Ladin et al. [43], telehealth has been posited as a cost-effective means for improving access to care for persons with chronic conditions, including kidney disease. Lyth et al. [34] show that the intervention through an active telemonitoring system was effective for both patients with heart failure and COPD in reducing hospitalizations of patients who died during the study period, as well as for patients who survived the entire study period. In addition, the intervention reduced the number of visits to primary care, emergency care, and other outpatient care.

### 3.3. Consultation

In the second focus group meeting to discuss the study findings and achieve consensus, it was agreed that the number of included articles was sufficient regarding the validity and scope of the research and that the findings were accurate. Given that older adults have different physical, emotional, and social needs, telecare services should not be standardized, and older adults should not be considered a homogeneous group.

In providing telecare services, the countries’ economic structures are as important as their socio-cultural structures. For example, some barriers to telecare use include income decline after retirement and the fact that older individuals may not afford the installation of a telecare system in their current housing conditions. For these reasons, the authors of this review call upon central and local governments to support telecare services.

## 4. Discussion

This review examined the research on telecare and old age. Forty articles were analyzed, their key themes were identified, trends were revealed, and research findings were discussed. Although the content of telecare services varies, studies have concluded that telecare refers to a range of services that use alarms, sensors, and other equipment to help individuals live independently. The benefits of using telecare at home include maintaining an independent life at home and ensuring security [7,23,28]. However, the key aspect here is the meaning of independence. Although telecare supports older individuals in leading an independent life at home, it does ensure that they will be fully independent. In fact, telecare may be seen as a threat to their independent identity, stigmatizing, and having the potential to exclude and lead to loneliness [20,58], due to the reduction of face-to-face care. Telecare research on older individuals focused mainly on the use, effectiveness, and efficiency of this service, without addressing the social and ethical issues involved [22,50].

Telecare options should be carefully assessed given their potential benefits and risks [58]. Older people’s low technology acceptance level and the complexity of telecare devices may deter them from using telecare. The current telecare designs are criticized for being unsuitable for older individuals [21,25,30,42]. Cameras used to monitor domestic life are considered both a system that ensures security and a threat to privacy [22]. On the one hand, it eliminates the need for family caregivers to spend all their time with older adults, and on the other hand, it may bring additional responsibilities to caregivers [23]. Telecare also poses major challenges for healthcare professionals as it requires both technical and clinical skills, but few studies address these challenges [1,45,47].

The incidence of health problems increases with age, with older people having to prolong their hospital stay or receive institutional care. Possible benefits of telecare include reducing maintenance costs and avoiding unnecessary admission to institutional care [49,59,60,61]. Telecare is considered critical in meeting the future care needs of aging societies [24]. While the expected outcome of telecare use is the reduction of care-related costs, this review shows that there are few studies on its cost-effectiveness and that the available studies suggest inconsistent results.

Chronic disease is the major cause of healthcare utilization worldwide. Its prevalence is increasing because the world’s population is aging. According to the study by Gao et al. [62], the aging of the population and the growing number of individuals with chronic diseases require more innovative and effective models of care delivery to improve older people’s quality of life. Chronic disease management in the home environment is a challenge, and telehealth care can be a vital help in monitoring the condition and in self-care [62,63].

On the other hand, a telecare support system can effectively improve dementia care [27,51]. The results show that the intervention using an active telemonitoring system was effective for both heart failure and COPD patients in reducing hospitalizations of patients who died during the study period, as well as for patients who survived the entire study period [34,53].

This scoping review only included studies on telecare and aging published in English. Due to this limitation, it may have left out studies published in languages with a potentially higher number of studies on technological services, such as those in Japanese, Chinese, and German.

## 5. Conclusions

Today’s care-related policies focus on care services that support aging in place and independent living at home. Telecare technological solutions are included in health services. Therefore, telecare should also be assessed considering the concepts of aging in place and active and independent aging, which have a prominent place in the gerontology literature. Studies on telecare need to focus on the extent to which it supports older people’s independence, engagement, and identity.

Further studies on the cost-effectiveness and ethics of telecare are needed, particularly given the recent technological advancements. In this respect, studies are needed to develop an ethical framework for social care and examine the biomedical ethical dimension in telecare use.

This scoping review revealed that the majority of studies on old age and telecare focus on service use and apply qualitative research methods to reveal the nature of the service and the care stakeholders’ perceptions. On the other hand, few studies are representative of this population. Combined methods should be used to attend to the different characteristics and needs of the target population.

It is difficult to compare the findings of studies on telecare across countries because each country has implemented different healthcare systems supporting telecare systems, and there is no international consensus on the telecare concept. Further research comparing telecare services across countries is needed to develop a shared concept of telecare.

## Figures and Tables

**Figure 1 ijerph-19-16672-f001:**
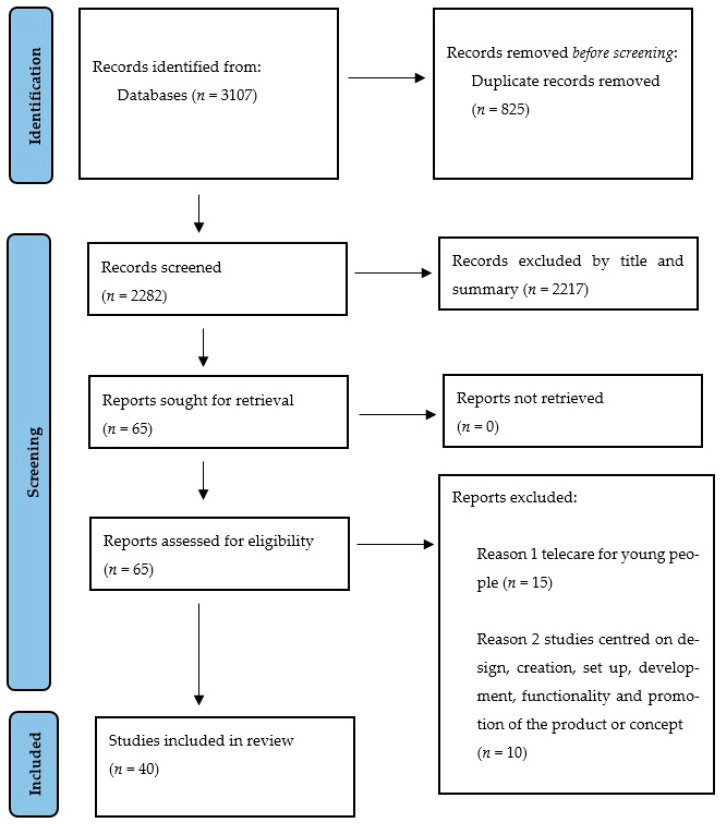
PRISMA (Preferred Reporting Items for Systematic Reviews and Meta-Analysis extension for scoping reviews [17]).

**Figure 2 ijerph-19-16672-f002:**
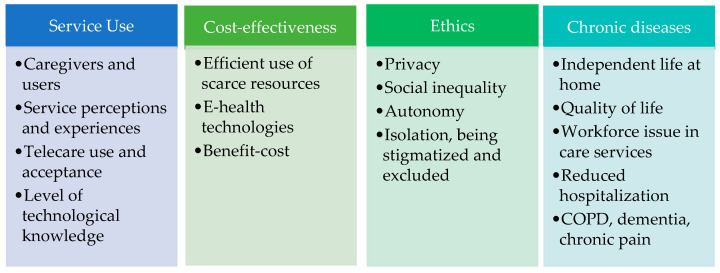
Key themes and sub-themes.

## Data Availability

For data supporting reported results please contact the authors of this review.

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
