# Peer review of "Trends in Telecare Use among Community-Dwelling Older Adults: A Scoping Review"

_ijerph, 2022, doi:10.3390/ijerph192416672_

Round 1
Reviewer 1 Report
The authors conducted a scoping review aiming to map and analyze the concept of telecare services and the trends in telecare use. Using a structured review process, the authors reviewed 40 published research articles. Through this review, the authors synthesized the current key themes, potential health benefits, and unsettled issues regarding telecare. The findings of this well-written scoping review are essential. There are some comments.
Comments:
1. Introduction: The review questions and objectives should be specified at the end of the Introduction.
2. Method (2.2. Identifying relevant studies) (Page 3): I would suggest specifying the date the most recent search was executed.
3. Method (2.4. Charting the data) (Page 4, Line 154-155): “Four independent reviewers extracted the data using an extraction tool developed specifically for this scoping review.” It is unclear whether each study was reviewed by four reviewers independently. I would suggest describing more the data charting (extraction) process conducted in this review. The information should include the following: the number of reviewers for each published study, whether they worked independently and compared answers or some researchers charted and other researchers verified, how inconsistencies or disagreements were resolved, (if applicable) any processes for obtaining and confirming data from investigators, etc.). In addition, more information about how the “extraction tool” was developed was needed. Specifically, it is unclear how the items were selected. Also, has calibration (testing the tool among research team members) of the “extraction tool” been conducted? If yes, the authors are advised to describe the details of the process (including the number of persons who tested the tool, how inconsistencies were resolved, and critical changes that were made and why).
4. Abstract: Eligibility criteria and charting methods should be described in the abstract.
Reviewer 2 Report
I would like to congratulate the authors for creating such a broad review of the literature in a field that is still developing. I think that is extremely necessary to systematize the existing knowledge about them.
Conclusions are rather descriptive but this is coherent with the purpose of presented review which to my understanding was presenting trends in telemedicine rather than rating their usefulness.
Issues needing correction are rather minor among them I would advocate enhancing the resolution of figure 1 on page 4 to make it more readable and adjusting the finish of table 1 on page 19 for the page not to look so empty.
Therefore I render this review suitable for publishing with no further corrections
